# A Quantum Double-or-Nothing Game: An Application of the Kelly Criterion to Spins

**DOI:** 10.3390/e26010066

**Published:** 2024-01-12

**Authors:** Bernhard K. Meister, Henry C. W. Price

**Affiliations:** 1FastEagle Holdings, 1200 Vienna, Austria; 2Centre for Complexity Science, Physics Department, Imperial College London, London SW7 2AZ, UK; henry.price10@imperial.ac.uk

**Keywords:** quantum games, quantum finance, derivatives, kelly criterion, portfolio theory

## Abstract

A quantum game is constructed from a sequence of independent and identically polarised spin-1/2 particles. Information about their possible polarisation is provided to a bettor, who can wager in successive double-or-nothing games on measurement outcomes. The choice at each stage is how much to bet and in which direction to measure the individual particles. The portfolio’s growth rate rises as the measurements are progressively adjusted in response to the accumulated information. Wealth is amassed through astute betting. The optimal classical strategy is called the Kelly criterion and plays a fundamental role in portfolio theory and consequently quantitative finance. The optimal quantum strategy is determined numerically and shown to differ from the classical strategy. This paper contributes to the development of quantum finance, as aspects of portfolio optimisation are extended to the quantum realm. Intriguing trade-offs between information gain and portfolio growth are described.

## 1. Introduction

An early application of information theory to gambling and finance can be found in a 1956 paper by John L. Kelly [1]—an associate of Claude Shannon. Kelly showed how a logarithmic utility maximising investor should allocate capital between bets with known winning probabilities and payouts. In the simplest case, the investor bets a fraction of wealth on the outcome of a biased coin in the ‘double-or-nothing’ game. An outcome of ‘heads’ doubles the gambler’s stake, while ‘tails’ implies that the stake is lost. The optimal fraction—called the ‘Kelly criterion’—is to bet 2p−1 for a coin that comes out heads with probability *p*, assuming p≥1/2. This maximises the logarithmic utility, which is equivalent to maximising the growth rate of the portfolio.

Kelly’s results are an elaboration of the earlier works of Daniel Bernoulli and others, who studied the ‘St. Petersburg paradox’ ( for a review of the history of the ‘St Petersburg paradox’, see the relevant page of the online Stanford Encyclopaedia of Philosophy: https://plato.stanford.edu/archives/fall2023/entries/paradox-stpetersburg/ accessed on 24 December 2023). The question was how to evaluate a coin-flipping game, where the payout depends on how often ‘heads’ occurs in succession. If ’tails’ occurs first on the *N*-th throw, then the payout is equal to $2^{N}$. The coin is considered ’fair’, and the probability for a having a payout $2^{N}$ is $2^{-N}$. The paradox was that whilst the expectation value of the bet is not finite, the value that gamblers were willing to pay for such a payout was finite. One ‘resolution’ of the paradox goes back to Daniel Bernoulli, who in 1738 suggested that large gains should be discounted more than smaller gains or losses, since gains and losses are perceived nonlinearly due to risk aversion. In general, returns should be adjusted for risk. As the saying goes, ‘a bird in the hand is better than two in the bush’. Bernoulli suggested looking not directly at the profit gained in the bet, but constructing instead a value function, which turns gains into utility. For this value function, he suggested the logarithm of the gain. This shift benefits from absolute to relative gains and makes the result independent of current wealth of the bettor. His derivation uses the equation $dy=k\frac{dx}{x}$, where *y* is value or utility, *x* is wealth and *k* is a constant. This leads to y=klogx+c. The result is that gambles are worth taking if the value or utility increases when participating in the gamble at the price asked. The utility of the bet following Bernoulli’s description is then
(1)$$\begin{array}{c} \sum_{i=1}^{\infty }2^{-i}\log\left(2^{i}\right)=\log4 \end{array}$$
assuming k=1 and c=0. Therefore, a bettor following Bernoulli’s rationale would be willing to pay up to four units to play the game. A person offering the bet would have to account for possible big losses, which are accounted differently from big gains and would not normally arrive at the same price.

The work of Kelly was championed by Cover, Ziemba and others, and found general application in finance. It overlaps with the popular mean-variance portfolio theory introduced by Harry Markowitz. A review of classic and modern developments of the Kelly criterion can be found in MacLean et al. [2]. Some germane properties of Kelly portfolios are described in [3,4].

Instead of a series of coin flips, we consider a stream of spin-1/2 particles that are measured sequentially. Again, there is a ‘double-or-nothing’ game to wager on, but the payouts depend on the results of variable measurements. What distinguishes the classical from the quantum case is the added degree of freedom associated with the measurement directions of the quantum particles. Instead of just flipping a coin, one can measure the spin particle in an arbitrary direction. The outcome probability is direction dependent.

This paper highlights some novel aspects of quantum game theory, quantum gambling and in extension quantum finance. ‘Quantum gambling’ is likely to have applications in finance as the quantum scale becomes of practical importance in communication and computation. The importance of hedging against errors that have economic consequences rises. The quantum gambling toy model described in the paper should be of relevance for analysing such situations. The conclusion provides further information.

The summary of the rest of the paper is as follows. The next section introduces gambling with spin-1/2 particles, followed by a section on the quantum version of the betting game. The subsequent section gives a heuristic description of the optimal strategies. Section 5 delves into numerical calculations. In a sub-section, the special case of prior 1/2 is considered. Section 6 covers the optimal strategy expressed as ‘contours of equivalence’, and Section 7 concludes the paper.

## 2. Gambling with Spin-1/2 Particles

This section describes how one can ‘quantum gamble’ with spin particles. A gambler, or to use the more courteous term, investor (W. F. “Blackie” Sherrod: “If you bet on a horse, that’s gambling. If you bet you can make three spades, that’s entertainment. If you bet cotton will go up three points, that’s business. See the difference?”), is presented with a sequence of quantum spin-1/2 particles. Unknown to the investor is the polarisation of the particles, which are either all prepared in state ρ or state σ. The investor is only informed that the prior probability for ρ is ξ and for σ is 1−ξ.

The investor is further told that a ‘double-or-nothing’ game is linked to measurements of the particles. The investor can bet any fraction below one of owned assets on the outcome of the measurement, i.e., leverage is not considered. If the measurement outcome in the direction of choice is spin-up, then the investor’s stake is doubled. If, on the other hand, the outcome is spin-down, the stake is forfeited.

To maximise the probability of winning a particular bet, an optimal measurement direction must be determined. If a sequence of bets is to be considered, another factor influences the choice, since judicious measurements are accompanied by information gain and allow progressively for more accurate determination of the polarisation direction of the particles. Balancing short-term winning probability with information gain and increased profitability is novel and reflects the quantum nature of the problem. In the next section, some relevant calculations for spin gambling are presented.

## 3. The ‘Double-or-Nothing’ Game with Spin-1/2 Particles

This section gives further details of the spin-1/2 gambling game. Notation conducive to the problem at hand is introduced next. The candidate states are |ψ〉=sinγ|0〉+cosγ|1〉 and |ϕ〉=sinβ|0〉+cosβ|1〉, more conveniently written as $|\phi \rangle =\cos\delta |\psi \rangle +\sin\delta |\psi_{⊥}\rangle$ with
(2)$$\begin{array}{c} \cos^{2}\delta ={\left(\sin\gamma \sin\beta +\cos\gamma \cos\beta \right)}^{2}=\cos^{2}\left(\gamma -\beta \right), \end{array}$$
and $|\psi_{⊥}\rangle =\sin\gamma |0\rangle -\cos\gamma |1\rangle$. To simplify, we set γ=0, and the remaining relevant angle is δ. The density matrices are ρ=|ϕ〉〈ϕ| and σ=|ψ〉〈ψ|, with the respective priors of ξ and 1−ξ. The derivation of the optimal informational strategy for the case of two polarisation directions can be found in Helstrom [5] and an extension to multiple particles is mentioned in Brody et al. [6]. If one considers additional polarisation directions, the optimal strategy has only been obtained numerically.

The tuneable inputs are then reduced to δ, ξ and *N*, where *N* corresponds to the number of particles targeted and measured out of a potentially larger stream. The index, *i*, runs from 1 to *N*. The initial wealth, $W_{1}$, is 1 without loss of generality since the initial wealth can be arbitrarily scaled. Later in the paper, we do not fix the initial but the final wealth. If a particular initial wealth is preferred, one can simply re-scale $W_{N}$. The optimal measurement angle to maximise the information gain is
(3)$$\begin{array}{c} \phi =\tan^{-1}\left(\frac{\left(\xi -1)\sin(\delta \right)}{\xi -(1-\xi )\cos(\delta )}\right), \end{array}$$
and the optimal portfolio growth angle, $\hat{\phi }$, is ϕ shifted by π/4, i.e., $\hat{\phi }=\phi +\pi /4$, if and only if ξ=1/2. Figure 1 shows the measurement directions in the case of equal prior binary state discrimination. The optimal measurement directions are adjusted for unequal priors. If the prior of the state, |ϕ〉, approaches one, then the optimal measurement directions will tend towards the orthogonal pair that is parallel and orthogonal to the state |ϕ〉 and lies in the plane spanned by the two candidate states. For a detailed account of the derivation of Equation (Equation 3) and the interpretation of the procedure, the reader should consult [6]. For each of the *N* particles gambled on and analysed, a separate $\alpha_{i}$ is chosen. The probability of a spin-up outcome for such a measurement in the $\alpha_{i}$ direction is as follows:(4)$$\begin{array}{c} p_{i+1}^{up}=\xi_{i}\cos^{2}\left(\alpha_{i}\right)+\left(1-\xi_{i}\right)\cos^{2}\left(\delta -\alpha_{i}\right). \end{array}$$ The updating of the prior for spin-up leads to the outcome of
(5)$$\begin{array}{c} \xi_{i+1}^{up}=\frac{\xi_{i}\cos^{2}\left(\alpha_{i}\right)}{\xi_{i}\cos^{2}\left(\alpha_{i}\right)+\left(1-\xi_{i}\right)\cos^{2}\left(\delta -\alpha_{i}\right)}, \end{array}$$
and for spin-down, the outcome is
(6)$$\begin{array}{c} \xi_{i+1}^{down}=\frac{\xi_{i}\left(1-\cos^{2}\left(\alpha_{i}\right)\right)}{\xi_{i}\left(1-\cos^{2}\left(\alpha_{i}\right)\right)+\left(1-\xi_{i}\right)\left(1-\cos^{2}\left(\delta -\alpha_{i}\right)\right)}. \end{array}$$ The optimal investment fraction for a chosen $\alpha_{i}$ is $f_{i}=2p_{i}^{up}-1$, if $p_{i}^{up}\geq 1/2$; otherwise $\alpha_{i}$ is shifted by π/2, replacing $p_{i}^{up}$ by $1-p_{i}^{up}$. The updating of the wealth process leads, in the case of a spin-up outcome, to
(7)$$\begin{array}{c} W_{i+1}=W_{i}\left(1+f_{i}\right), \end{array}$$
and, in the case of a spin-down outcome, it leads to
(8)$$\begin{array}{c} W_{i+1}=W_{i}\left(1-f_{i}\right). \end{array}$$ If now one simulates a chain of measurements in the $\alpha_{i}$ directions, then one obtains a series of spin-up and spin-down outcomes with probabilities $p_{i}^{up}$ and $1-p_{i}^{up}$, respectively. Let us represent a spin-up outcome by + and a spin-down outcome by −. The outcome sequence would then look like +−+++−…+−++++++, with progressively more +’s as one edges the prior away from 1/2. The optimisation algorithm will adjust $\alpha_{i}$ to maximise the probability-weighted logarithmic utility of $W_{N}$:(9)$$\begin{array}{c} \max_{all\;\;\;possible\;\;\;\alpha_{i}\;\;\;sequences}\sum_{all\;\;\;the\;\;\;2^{N}\;\;\;+/-\;\;\;sequences}p\left(\ldots \right)\log W_{N}\left(\ldots \right). \end{array}$$ The different outcome sequences have probabilities p(…), i.e., all the way from p(+…+) to p(−…−), associated with them. Their probability is a product of *N* terms of the form $p_{i}^{up}$’s and $p_{i}^{down}=1-p_{i}^{up}$, i.e., *i* running from 1 to *N*. A section on the optimal strategy is presented next.

## 4. The Optimal Strategy: A Heuristic Description

In this section, a heuristic description of the optimal strategy is given. It requires the backward solution of an optimisation problem, similar to the evaluation of a European option on a binomial tree. As background, a European option is a contingent claim that gives the owner the right but not the obligation to buy or sell an agreed asset at an agreed option termination time for a fixed price—the so called ‘strike price’. The value at the expiry date lies for call and put options in the difference between the market price and the strike price. In the case of call options, the profit is derived from the right to buy an asset at the strike price when the market price is higher; for put options, the profit is derived from the right to sell at the strike price when the market price is lower. Many other types of contingent claims exist, and in particular since the advent of the Black–Scholes option pricing formulae in the seventies these products and other derivatives have driven to a large extent the expansion of the financial industry. For the backward solution of the optimisation problem the values are calculated for all cases at the terminal time. The value function can then be evaluated for the previous step since the optimal earlier values are the maximum over the available decision space of the probability weighted averages of later values. The choice of the right probability measure requires some care. In our case, we assume that the whole problem is in what finance theorists call the ’risk neutral’ setting such that no adjustment of the probability measure is necessary. The backward calculation can be carried out iteratively all the way to the first step.

If only one particle remains to be measured, then maximisation of the spin-up probability is key, since any ‘information gain’ cannot be exploited in the future. Therefore, one solely maximises portfolio growth in the last round. This determines the angle $\alpha_{N}$ and with it the spin-up probability, i.e.,
(10)$$\begin{array}{ccc} 0 & = & \partial_{\alpha_{N}}(\xi \cos^{2}\left(\alpha_{N}\right)+\left(1-\xi \right)\cos^{2}\left(\alpha_{N}-\delta \right))\\ 0 & = & \xi \cos\left(\alpha_{N}\right)\sin\left(\alpha_{N}\right)+\left(1-\xi \right)\cos\left(\alpha_{N}-\delta \right)\sin\left(\alpha_{N}-\delta \right)\\ 0 & = & \tan^{2}\left(\alpha_{N}\right)\sin\left(\delta \right)\cos\left(\delta \right)+\tan\left(\alpha_{N}\right)\left(\frac{\xi }{1-\xi }+1-2\sin^{2}\left(\delta \right)\right)-\sin\left(\delta \right)\cos\left(\delta \right)\\ & & ⟹\alpha_{N}=\tan^{-1}\left(-\frac{A}{2}\pm \frac{1}{2}\sqrt{A^{2}+4}\;\;\right) \end{array}$$
with $A=\frac{\frac{\xi }{1-\xi }+1-2\sin^{2}\left(\delta \right)}{\sin(\delta )\cos(\delta )}$. The ‘Kelly criterion’ then allows for the determination of the optimal fraction, and the resulting wealth (and ξ) can be calculated for both measurement outcomes and with it the achievable probability-weighted utility.

Next, we move one step back. The angle $\alpha_{N-1}$ at step N−1 updates in a distinct way for the two possible measurement outcomes both the wealth and the prior. By maximising over the range of allowed measurement angles $\alpha_{N-1}$, one can determine the maximal achievable utility as the probability-weighted sum of the utility at the final step. This is then the utility associated with that particular point in the two-dimensional wealth and ξ space. A similar mechanism works for earlier steps all the way back to step one.

The optimal strategy is then given by the contour map of equivalent utility lines at each step over the wealth and ξ space. Figure 2, Figure 3, Figure 4, Figure 5, Figure 6, Figure 7, Figure 8 and Figure 9 show examples of contour lines for the polarisation angle δ of $7.5^{\circ },\;\;30^{\circ },\;60^{\circ }$ and $90^{\circ }$. Each angle comes with a pair of plots, i.e., with and without the wealth presented on a logarithmic scale. The points connected by the *k*-th contour line yield the same final wealth utility at step *N*. Any logarithmic utility maximising bettor should be indifferent between any of the points on these lines. The endpoints of the *k*-th contour line correspond to ξ=1 or ξ=0 and wealth of $W_{N}/\left(2(N-k\right))$, i.e., doubling of wealth until one reaches $W_{N}$. The other points on the same contour line have higher wealth but a ξ closer to 1/2, and therefore lower optimal winning probability. Besides the contour graphs, the heat maps (see Figure 10 and Figure 11) are also of interest, which show that utility is strongly linked to the remaining number of steps as well as wealth, *W*, but less to the prior, ξ. Next, a description of the optimal strategy, derived by numerical means, is presented.

## 5. Numerical Simulation: Algorithm and Pseudo-Code

The numerical simulation is discussed in this section. The Algorithm 1 works backwards and defines the utility surface at each step as a function dependent on the wealth value *W* and ξ. At the final step, *N*, all points on the straight-line contour have a fixed wealth, $W_{N}$, but an arbitrary value for ξ. This can be represented as a vector $\left(W_{N},\xi_{N},N,\log_{2}\left(W_{N}\right)\right)$. The utility surface for the previous step is then defined, and general equations for calculating $\left(U_{k}\left(W,\xi \right)\right)$ are given. The utility value for each step is calculated through an iterative process. The pseudo-code outlines the computational method for calculating the utility surface. The code results in a multidimensional array of utility values, with non-grid-point utilities evaluated by interpolation.
**Algorithm 1** Quantum Kelly Optimisation across N steps1:Define $\alpha_{\mathrm{range}}$ from 0 to $\frac{\pi }{2}$2:Define $W_{\mathrm{range}}$ with logarithmically spaced values between 0 and 23:Define $\xi_{\mathrm{range}}$ with logarithmically spaced values between 0 and 0.54:Define δ (adjustable parameter)5:Initialize utility array *U* with zeros6:Compute $U\left[N-1,:,:\right]=\log_{2}\left(W_{\mathrm{range}}\right)$         ▹ Initial value using log27:**for** k=N−2 to 0 **do**8:    Create interpolation for current stage based on $W_{\mathrm{range}}$ and $\xi_{\mathrm{range}}$9:    **for** each *W* in $W_{\mathrm{range}}$ **do**10:        **for** each ξ in $\xi_{\mathrm{range}}$ **do**11:           Compute $p_{+}$, $p_{-}$ based on ξ, α, δ12:           Compute $\xi_{+}$, $\xi_{-}$ based on ξ, α, δ13:           Compute $W_{+}$, $W_{-}$ based on *W*, $p_{+}$14:           Interpolate values $U_{+}$, $U_{-}$ for given $W_{+}$, $\xi_{+}$, $W_{-}$, $\xi_{-}$15:           Define utility function using the above values16:           Optimize α to find $\alpha_{★}$ that maximizes utility17:           Update U[k,i,j] with optimized utility value  ▹ Iterative update of U using probability weighted sum of previous step utilities18:        **end for**19:    **end for**20:**end for**

The visualisation of the result is described next. Upon evaluating the array of points, the algorithm extracts contours.

In summary, the described optimisation algorithm calculates a utility surface in a space characterised by both the wealth and the prior; by working backwards through iterative calculations, the algorithm can generate a contour plot visualising the utility values across the specified space.

The simulation was written in Python; NumPy was utilised for numerical operations; SciPy optimized and interpolated the modules; and Matplotlib was employed for visualisation.

## 6. Contours of Equivalence

In this section, a technique for visualising the optimal gambling strategy is introduced. It relies on finding points in the two dimensional wealth and prior space that lead to the same final ‘utility’ outcome. These sets of equal ‘utility’ points form contour lines. Each line differs by the number of remaining particles to be measured and consequently bets to be made.

All these contour lines have easily calculated levels of wealth at the *k*-th step from the end for extreme values of the prior. These anchor points for the prior, ξ, at 0 and 1 are associated with a wealth of $2^{k-N}W_{N}$. There is a simple strategy that moves from these points to the final contour. Since the probability of winning is one, the gambler bets everything and doubles the money at every stage. Between these two extremes lies a range of points, which require numerical evaluation.

### The Curious Case of Growth without Information Gain or Information Gain without Growth

In this subsection, the special case of prior one-half is considered. If the prior, ξ, is set equal to 1/2, formulas simplify, and a curious situation arises. The optimal angle for information gain is described above, while a measurement achieves the one-round maximal growth in a shifted direction. These two directions are rotated by π/4; the maximal information gain is accompanied by betting zero, whereas the maximal growth measurement direction leaves ξ unchanged at the value of 1/2. In the case of ξ equal 1/2, the two strategies are maximally incompatible, which is not the case for other values of ξ.

## 7. Conclusions

This paper investigated a quantum betting game and showed how an optimal strategy can be established. Some differences between the classical and quantum case were discussed, and some strategies were compared. Simple extremal strategies were most easy to compare. One could, for example, maximise ‘information gain’ or ‘portfolio growth’. As we have shown, the optimal strategy lies somewhere in between and depends on δ, ξ and the number of steps *N*. If ξ is either zero or one, then information gain is impossible and maximising short and long-term growth becomes synonymous. A special case is when ξ equals one half, where an optimal ‘information gain’ measurement entails zero ‘portfolio growth’ and optimal ‘portfolio growth’ entails zero ‘information gain’.

One serious challenge not addressed in this paper is how to implement such a game in a laboratory where there are measurement errors and delays in the analysis of data, while particles in the beam stream past could interfere with or even wreck the optimal strategy. This raises issues that depend on the particular implementation of the game and are beyond the scope of this paper. One of the consequences of measurement errors can be an over- or underestimation of the optimal betting amount. The knowledge of even unbiased measurement errors in particular leads to a more conservative betting strategy, since the downside risk of over-betting is larger than the potential lost gain from under-betting due to asymmetric risk in the Kelly criterion, as explained in the paper by Thorpe [7].

Currently, ‘quantum gambling’ is not widely applied in the real world and limited to the laboratories or the domain of finance. However, besides the ability to probe and test quantum mechanics, one could envisage scenarios where the outcomes of quantum-scale events play a role in decision making. In this case, quantum game theory and applications like the quantum gambling problem discussed in the paper are of interest.

One can further find applications in quantum computing and information theory. Imagine a quantum communication channel, where a misinterpretation of the message leads to a financial loss. The evaluation and insurance against such faults can be formulated in certain cases as a quantum gambling problem of the type discussed in this paper. 

## Figures and Tables

**Figure 1 entropy-26-00066-f001:**
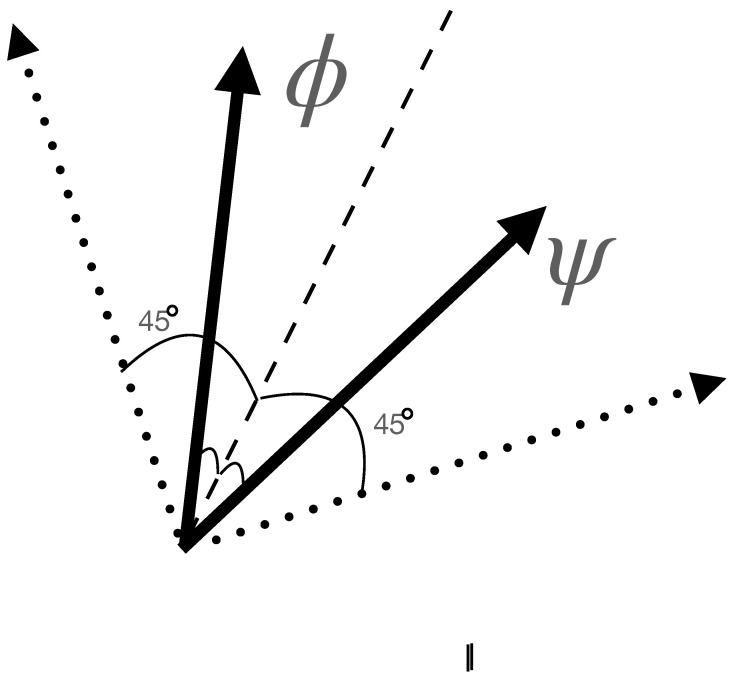
The plot shows optimal measurement directions as dotted vectors for distinguishing equal prior states |ϕ〉 and |ψ〉 represented as solid vectors. The dashed line equally divides the angle between the two states. The two dotted vectors are orthogonal, with one rotated clockwise by $45^{\circ }$ and the other one rotated counter-clockwise by the same angle from the dashed line.

**Figure 2 entropy-26-00066-f002:**
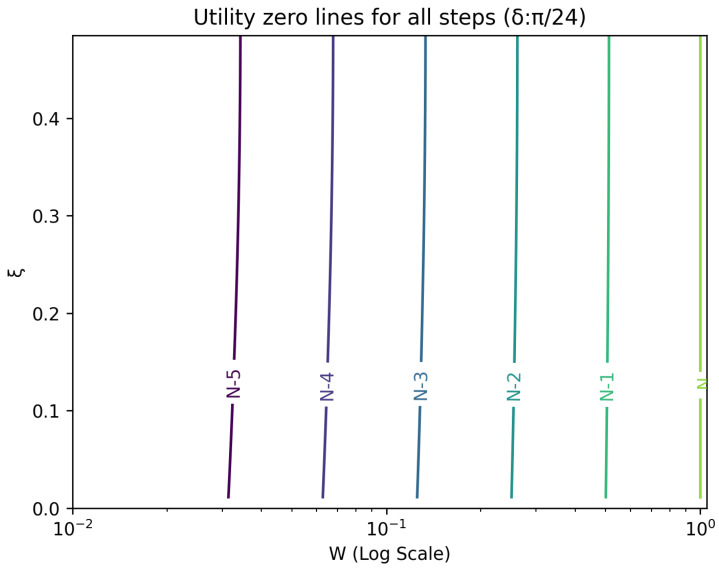
Plot of contour lines for same utility at different steps for $\delta =7.5^{\circ }$ using a logarithmic scale on the x-axis for *W*.

**Figure 3 entropy-26-00066-f003:**
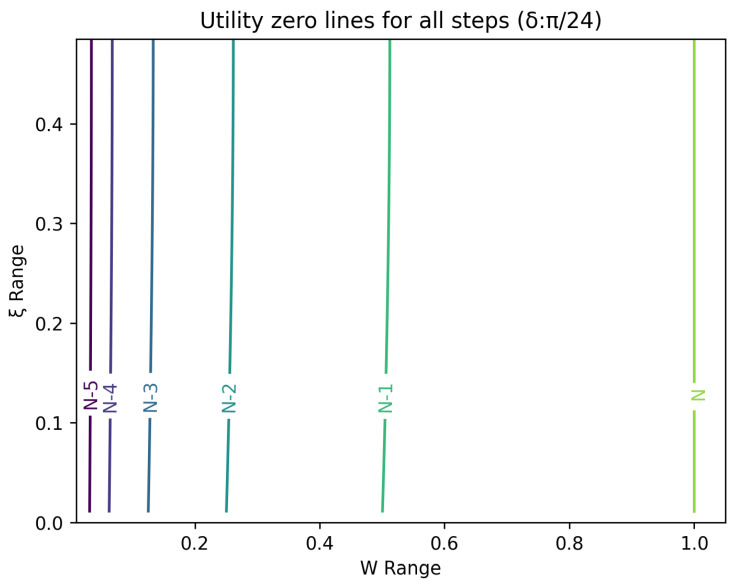
Plot of contour lines for same utility at different steps for $\delta =7.5^{\circ }$.

**Figure 4 entropy-26-00066-f004:**
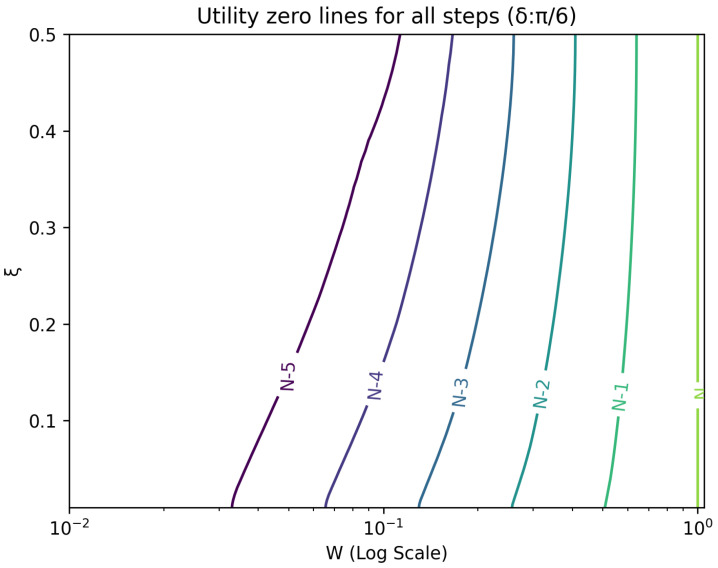
Plot of contour lines for same utility at different steps for $\delta =30^{\circ }$ using a logarithmic scale on the x-axis for *W*.

**Figure 5 entropy-26-00066-f005:**
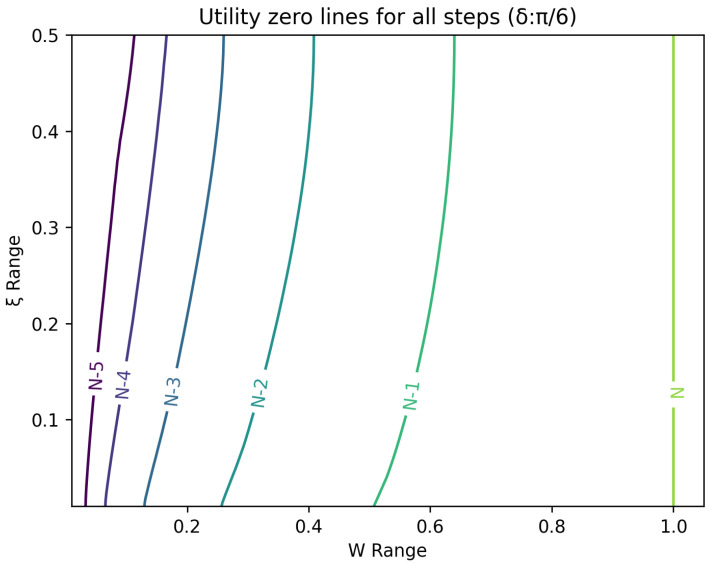
Plot of contour lines for same utility at different steps for $\delta =30^{\circ }$.

**Figure 6 entropy-26-00066-f006:**
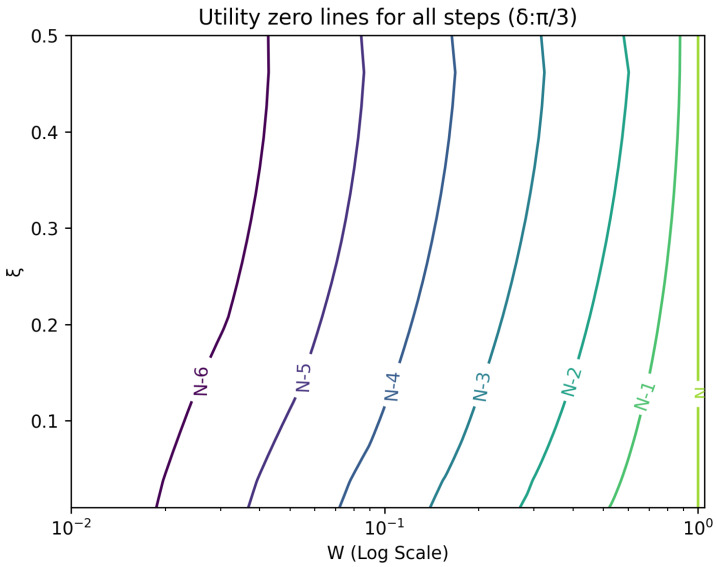
Plot of contour lines for same utility at different steps for $\delta =60^{\circ }$ using a logarithmic scale on the x-axis for *W*.

**Figure 7 entropy-26-00066-f007:**
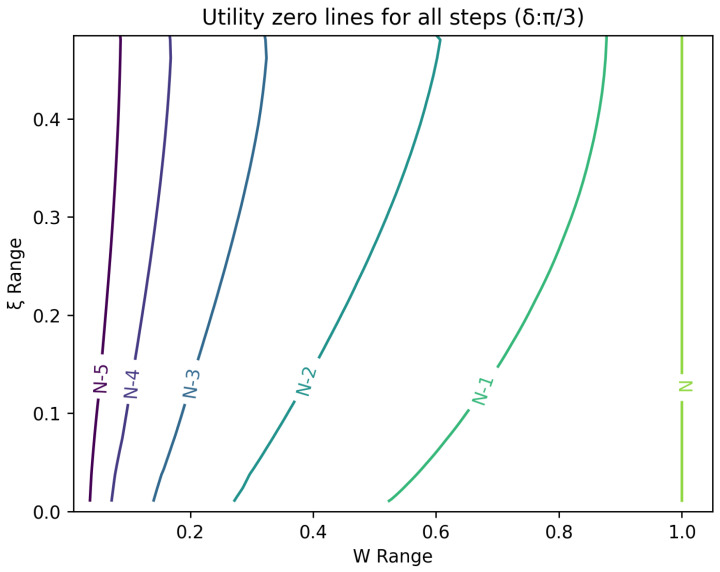
Plot of contour lines for same utility at different steps for $\delta =60^{\circ }$.

**Figure 8 entropy-26-00066-f008:**
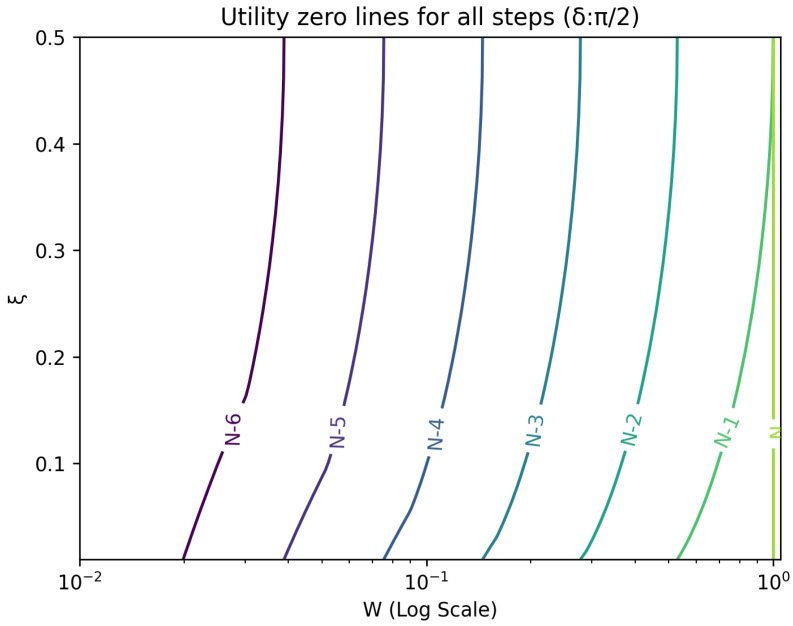
Plot of contour lines for same utility at different steps for $\delta =90^{\circ }$ using a logarithmic scale on the x-axis.

**Figure 9 entropy-26-00066-f009:**
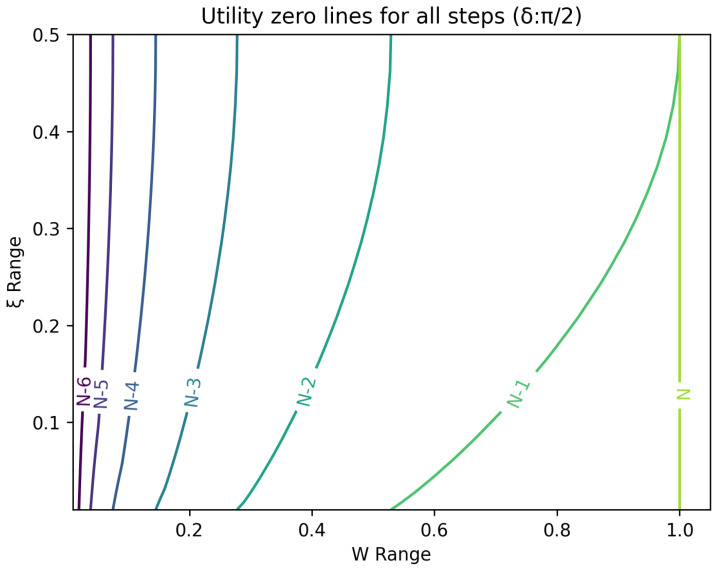
Plot of contour lines for same utility at different steps for $\delta =90^{\circ }$ using a standard scale on the x-axis.

**Figure 10 entropy-26-00066-f010:**
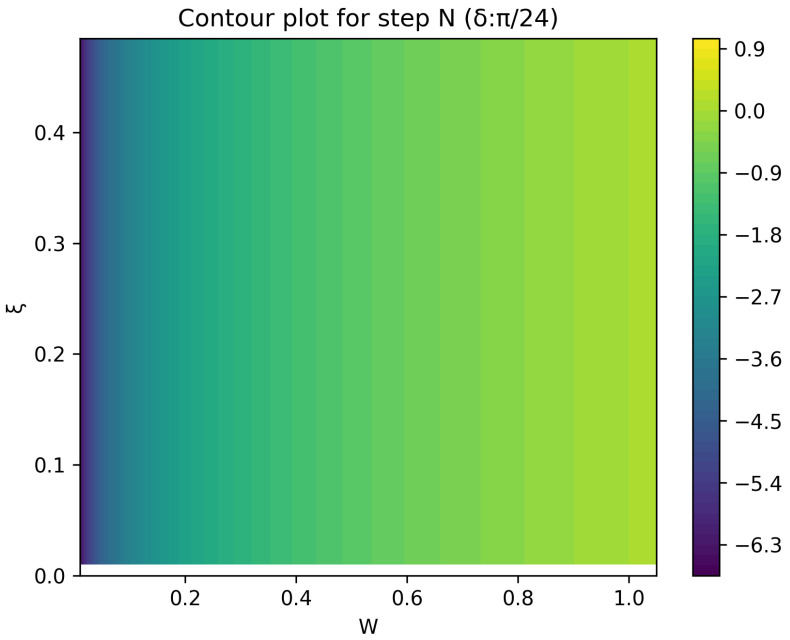
Heat map of utility at the final step.

**Figure 11 entropy-26-00066-f011:**
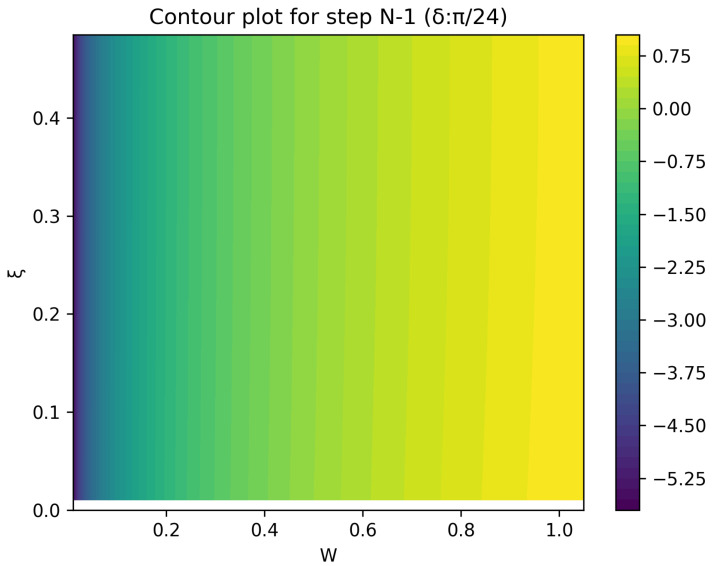
Heat map of utility at the second to final step.

## Data Availability

No new data were created or analysed in this study. Data sharing is not applicable to this article.

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
