# Peer review of "A Quantum Double-or-Nothing Game: An Application of the Kelly Criterion to Spins"

_entropy, 2024, doi:10.3390/e26010066_

Round 1
Reviewer 1 Report
Comments and Suggestions for Authors
The authors analyze the double-spin game and demonstrate certain optimal strategies based on the delta angle. My impression is that this is an interesting article with novel results and could be publishable, but I find it challenging to follow the logic of the mathematical procedure. In fact, I had to carefully read reference [6] to understand most of the mathematical procedure. It would be very helpful for the reader if figures were included in the development to help understand how the inverter interacts with the Stern-Gerlach apparatus.
Author Response
Thank you for the comments and stating that there is a necessity to connect more directly binary quantum state discrimination as given in reference [6] to our paper.
The methodology of [6], which is co-authored by one of the authors of the current paper, is now linked more closely, see lines 101-106 in section 3. The text emphasises the necessity to consult the earlier paper for the derivation needed in the current paper. The paper under review is built on the earlier results and can be viewed as an application of the earlier research.
A graph (figure 1) has been added to show in the equal prior case the optimal measurement directions.
Reviewer 2 Report
Comments and Suggestions for Authors
The authors present a quantum version of the Kelly Criterion utilizing a stream of polarized spin-1/2 particles. The work is interesting and original: I do not know of any previous quantum versions of this situation and the paper has little relation to other published models of quantum gambling.
I have some suggestions for improving the paper which I note as comments on the PDF. In general, equations should be numbered and references added when quoting known results, e.g. the equation on line 96. On line 117, it is necessary to explain how a European option works - most readers in quantum physics will not be familiar with this. In addition, I note a couple of typographic errors in the comments. With these minor improvements I am happy to see the paper published.

I note a couple of typographic errors and places clarity can be improved but overall the paper is clear.
Author Response
Thank you for the helpful comments, which have been incorporated.
The superfluous footnote mentioning the play ‘Rosencrantz and Guildenstern’ has been taken out.
There are now lengthy footnotes on backward optimisation and European options.
The optimal angle calculation (just before line 101) has now been linked explicitly (see lines 101-106) to reference [6] and supplemented by Figure one.
As stated above in comments to the first referee this paper can be viewed as an application of the earlier paper [6]. Except for the optimal angle derivation, the rest of the quantum mechanical calculations of the current paper are self-contained.
The 2017 paper by Kaminski is ingenious. We consider instead a single player game (or player vs nature), where the action of the player does not change spin measurements in the future (independence assumption). We therefore don’t see a direct application of the ideas discussed by Kaminski. This is at least our initial conclusion. By the way, if the particles are not independent (but entangled), then one measurement could interfere with future measurements and the situation changes. Thank you for the reference!
Reviewer 3 Report
Comments and Suggestions for Authors
The work is interesting and deserves publication. However, I have a few suggestions to improve the quality of the presentation of the results. The basis of the considerations is the particles' quantum behavior, so the experiment's description cannot ignore it. Therefore, I suggest that:
1. In the first line (Abstract) replace "particles polarized in one of two possible directions" with the simpler and more correct "polarized particles".
2. line 70: Could "probability" be replaced with "prior probability"? The authors call this parameter probability and prior interchangeably.
3. line 93 ("the number of available particles N"). From a physical point of view, it is unlikely that it would be possible to successfully implement such a system where exactly N particles were used and each could be "targeted". A more realistic situation is when we have a stream of polarized particles and make N measurements on them. In this context, it should be clearly noted that we do not change the polarization of this stream.
Moreover, which is quite non-trivial, the implementation of such a game will always be burdened with errors in the preparation of the particle stream and errors in measuring their polarization - the authors should comment on this problem, e.g. by describing how they imagine the implementation of such a game.
Author Response
Thank you for the comments!
>1. In the first line (Abstract) replace "particles polarized in one of two possible directions" with the simpler and more >correct "polarized particles".
The abstract has been rewritten.
>2. line 70: Could "probability" be replaced with "prior probability"? The authors call this parameter probability and prior >interchangeably.
Yes, done.
>3. line 93 ("the number of available particles N"). From a physical point of view, it is unlikely that it would be possible to >successfully implement such a system where exactly N particles were used and each could be "targeted". A more >realistic situation is when we have a stream of polarized particles and make N measurements on them. In this context, it >should be clearly noted that we do not change the polarization of this stream.
Thank you for the suggestion. The idea of picking particles from a stream has been included.
>Moreover, which is quite non-trivial, the implementation of such a game will always be burdened with errors in the >preparation of the particle stream and errors in measuring their polarization - the authors should comment on this >problem, e.g. by describing how they imagine the implementation of such a game.
Possible implementations and applications could come in many different guises as we say in the conclusion and a detailed evaluation is beyond the scope of the paper. I think in general, reformulating decision problems as games with well defined pay-out and optimal strategies is a useful exercise.
Errors are inevitable in implementations and a comment has been added about their effect on the optimal investment ratio. Due to the asymmetry of the logarithm chosen as the value function, i.e. betting zero leaves the utility unchanged, whereas betting too much can lead to bankruptcy ( log 0 = -infinity ), even unbiased errors lead to a more cautious approach.